# Unappreciated subcontinental admixture in Europeans and European Americans and implications for genetic epidemiology studies

Mateus H. Gouveia [1], Amy R. Bentley [1], Thiago P. Leal[2], Eduardo Tarazona-Santos[3], Carlos D. Bustamante[4], Adebowale A. Adeyemo [1], Charles N. Rotimi [1] ✉ & Daniel Shriner [1] ✉

European-ancestry populations are recognized as stratified but not as admixed, implying that residual confounding by locus-specific ancestry can affect studies of association, polygenic adaptation, and polygenic risk scores. We integrate individual-level genome-wide data from ~19,000 European-ancestry individuals across 79 European populations and five European American cohorts. We generate a new reference panel that captures ancestral diversity missed by both the 1000 Genomes and Human Genome Diversity Projects. Both Europeans and European Americans are admixed at the sub-continental level, with admixture dates differing among subgroups of European Americans. After adjustment for both genome-wide and locus-specific ancestry, associations between a highly differentiated variant in *LCT* (rs4988235) and height or LDL-cholesterol were confirmed to be false positives whereas the association between *LCT* and body mass index was genuine. We provide formal evidence of subcontinental admixture in individuals with European ancestry, which, if not properly accounted for, can produce spurious results in genetic epidemiology studies.

Human genetic studies have primarily considered admixed populations to have resulted from interbreeding between two or more continentally separated populations[1–3]. However, continental ancestry is not necessarily a single homogenous component of genetic diversity, but rather can be a composite of diverse subcontinental ancestries[4,5]. In some instances, differentiation between intra-continental populations is on par with or higher than differentiation between inter-continental populations[1,6]. Also, there are examples from pharmacogenetics of variants that are differentiated at the intra-continental level, such as in the case of abacavir hypersensitivity syndrome, for

which the causal allele (HLA-B*5701) has a prevalence of 13.6% among Maasai in Kenya but a prevalence of ~0% among Yoruba in Nigeria[7].

Despite genetic studies highlighting a clear pattern of North-to-South genetic variation in Europe[8–10] and strong evidence of admixture within Europe by ancient DNA analysis[11,12], European-ancestry populations are generally treated in association models as stratified but not as admixed at the subcontinental level. As a result, genetic epidemiology studies of Europeans or European Americans usually control for potential confounding effects of population stratification using genome-wide ancestry estimated by principal components analysis[13],

[1]Center for Research on Genomics and Global Health, National Human Genome Research Institute, National Institutes of Health, Bethesda, MD 20892, USA. [2]Department of Genomic Medicine, Lerner Research Institute, Cleveland Clinic, Cleveland, OH 44197, USA. [3]Departamento de Genética, Ecologia e Evolução, Instituto de Ciências Biológicas, Universidade Federal de Minas Gerais, Belo Horizonte, Minas Gerais 31270-910, Brazil. [4]Center for Computational, Evolutionary and Human Genomics (CEHG), Stanford University, Stanford, CA 94305, USA. ✉e-mail: rotimic@mail.nih.gov; shrinerda@mail.nih.gov

but do not control for locus-specific ancestry, which is inherent to admixed populations[14]. Potential consequences are that detection of causal genetic variation is hampered and estimation of effect sizes can be biased, leading to further negative consequences such as mis-estimation of polygenic adaptation[15] and poor predictive performance of polygenic risk scores[16].

Recently developed approaches have enabled the use of genome-wide data (either array-based genotype or whole genome sequence data) to assess admixture at two levels: genome-wide ancestry (also known as global ancestry)[13,17,18], which is the individual's ancestry averaged across the entire genome, and locus-specific ancestry (also known as local ancestry)[19–21], which allows for inference of an individual's ancestry at each locus. The power, resolution, and specificity of disease or trait mapping studies can be improved by leveraging both genome-wide and locus-specific ancestries[3,22,23]. To assess both genome-wide and locus-specific ancestries in admixed individuals, present-day populations are used as proxies for ancestral populations that serve as references for ancestry estimation. Considering that ~96% of participants in genome-wide association studies (GWAS) have European ancestry[24], a comprehensive analysis is needed to evaluate the adequacy of European reference panels for ancestry analysis using European-ancestry individuals.

The prevalence of lactase persistence varies widely across Europe and the most strongly associated variant rs4988235 in the lactase gene (*LCT*) has been reported to be under positive selection and associated with height, body mass index (BMI), and low-density lipoprotein (LDL)[25–28]. The SNP rs4988235 is one of the most highly differentiated variants in Europe[29], with derived allele (A) frequencies ranging from 93.1% in Swedes to 2.9% in Sardinians[30]. Importantly, rs4988235 and height are well known to covary following a north-to-south axis[31], and the association between rs4988235 and height has been suggested to be spurious based on attenuation following adjustment for genome-wide ancestry[27]. Nonetheless, there are no association studies in European-ancestry populations that control for confounding at both the genome-wide and locus-specific ancestry levels to test the validity of the association between rs4988235 and reported associated traits.

To test for the existence of subcontinental ancestries within Europe, we integrated genome-wide data from 1,216 individuals across 79 European populations. Then, to examine population structure and admixture, we integrated genome-wide data from 17,669 European Americans from five genetic epidemiology cohorts in the US. Finally, to illustrate the potential implications of confounding by subcontinental ancestry and admixture, we interrogated the association between rs4988235 and height, LDL-cholesterol, and BMI.

We found that the 1000 Genomes and Human Genome Diversity Projects provided incomplete coverage of European ancestries, so we generated a new reference panel to capture additional European ancestral diversity. Our admixture analyses yielded formal evidence that European-ancestry individuals are admixed at the subcontinental level, with admixture dates differing among European American subgroups. After adjustment for both genome-wide and locus-specific ancestry, previously reported associations between rs4988235 and height or LDL were no longer statistically significant, strongly supporting that they are false positives due to uncorrected stratification. We observed that better fits can be obtained when models were adjusted for principal components (PCs) derived from projection of European Americans onto our new reference panel, rather than for PCs derived from study-specific unsupervised analysis. Altogether, this study indicates that full adjustment for subcontinental European admixture (at both genome-wide and locus-specific levels) should become best practice in genetic association studies using European-ancestry individuals, including the UK Biobank[32] in Europe and the All of Us Research Program[33] and the VA Million Veteran Program[34] in the United States.

## Results

### Reference panels of European diversity

We generated a new reference panel capturing genetic diversity from 79 European populations from five population genetics studies: the 1000 Genomes Project[35], the Human Genome Diversity Project (HGDP)[36], the Human Origins dataset[37], a study of the Caucasus Mountains[38], and a study of the Jewish Diaspora[39] (Fig. 1A and Supplementary Data 1). After quality control to reduce batch effects, our European panel included 1,216 unrelated individuals and 104,414 genotyped SNPs. Principal component analysis (PCA)[13] showed that North Europeans (e.g., Finnish, Lithuanian, and Estonian) vs Southeast Europeans (e.g., Armenian, Georgian Jew, and Georgian Megrel) represented the extremes along the first principal component (Fig. 1B). Along the second principal component, Southwest Europeans (e.g., Sardinian, Basque, and Spanish) vs Southeast Europeans (e.g., South Caucasus) represented the extremes. Subsequent principal components separated population-specific genetic variability (Fig. S1). To compare our panel with commonly used European reference panels from the Human Genome Diversity Project (HGDP)[36] and the 1000 Genomes Project[35,36], we calculated convex hull areas[40] defined by the first two principal components (Fig. 1B, C), which captured 1.06% of the genetic variance. Compared to our panel, the 1000 Genomes Project and the HGDP covered 26.8% and 61.3% of European diversity, respectively, while the combination of the 1000 Genomes Project and the HGDP covered 77.3% (Fig. 1C). These results indicate that the 1000 Genomes Project and the HGDP, separately and combined, provide incomplete coverage of European genetic diversity.

### Subcontinental stratification in individuals with European ancestry

To expand and refine our understanding of subcontinental stratification and admixture in European-ancestry populations, we integrated genome-wide genotype data from approximately 19,000 European-ancestry individuals (Fig. 2). These data included our European panel (1216 unrelated individuals) and 17,669 European Americans from five genetic epidemiology cohorts in the US: Atherosclerosis Risk in Communities (ARIC, $n = 9633$), Coronary Artery Risk Development in Young Adults (CARDIA, $n = 1675$), Framingham Heart Study (FHS, $n = 2451$), Genetic Epidemiology Network of Arteriopathy (GENOA, $n = 1384$), and Multi-Ethnic Study of Atherosclerosis (MESA, $n = 2526$). To assess continental-level structure, we evaluated our European-ancestry dataset with a worldwide reference panel (Fig. S2). Most Europeans formed a discrete cluster along the first two principal components, as previously observed[35,36]. Similarly, by projecting European Americans (Supplementary Data 2) onto the worldwide reference panel, we observed that >99% of European Americans clustered with European reference individuals, with few individuals distributed along the first principal component (European-African gradient) or the second principal component (European-Asian gradient). These results suggest that the Europeans included in our panel represent a cluster in relation to worldwide genetic diversity and that European Americans in genetic epidemiology cohorts in the US have small to negligible population stratification at the inter-continental scale.

Next, to evaluate European subcontinental stratification in European American cohorts, we projected individuals from each European American cohort onto our European reference panel, represented by the first two principal components. We calculated that European American cohorts collectively covered 68.2% of European diversity in our panel (Fig. 2), with differential coverage by cohort: 55.7% in MESA, 51.2% in ARIC, 44.1% in CARDIA, 28.4% in FHS, and 9.7% in GENOA. The ARIC, CARDIA, FHS, and MESA individuals formed at least three clusters: one with North Europeans (e.g., British and Scandinavian), one with Southeast Europeans (e.g., Ashkenazi Jew and Romanian Jew), and one overlapping Finnish individuals. GENOA individuals mostly overlapped British or Scandinavian reference

individuals, with few individuals overlapping South Europeans. Most FHS samples overlapped with or were between North and South Europeans, with a large number of individuals clustering with Italian reference individuals. Most European Americans clustering with Finnish reference samples were from the ARIC cohort.

## Subcontinental admixture in individuals with European ancestry

Unsupervised analysis with ADMIXTURE[17] using our European reference panel identified the most likely number of ancestry clusters as three (Fig. 3A), suggesting that Europeans have three-way admixture among North, Southwest, and Southeast Europeans. The stacked bar plot of mixture proportions showed that the North European-associated ancestry cluster decreased along the north-to-south

geographic direction (Fig. 3A). Formal correlation tests between population ancestry means and geographic coordinates revealed that latitude was significantly correlated ($p < 2.85 \times 10^{-8}$) with the North European-associated ancestry cluster (Spearman's $rho = 0.814$), and longitude was correlated with Southwest- (Spearman's $rho = -0.859$) and Southeast-associated (Spearman's $rho = 0.579$) European ancestry clusters (Fig. 3B). We observed similar levels of genetic differentiation ($F_{ST}$) between the inferred European ancestry clusters: $F_{ST} = 0.033$ between North and Southwest, $F_{ST} = 0.032$ between North and Southeast, and $F_{ST} = 0.028$ between Southwest and Southeast. For these comparisons, European-associated ancestry clusters are genetically homogeneous populations identified by ADMIXTURE, not real-world populations. To put these amounts of genetic differentiation into perspective, $F_{ST}$ estimates between European ancestry clusters are

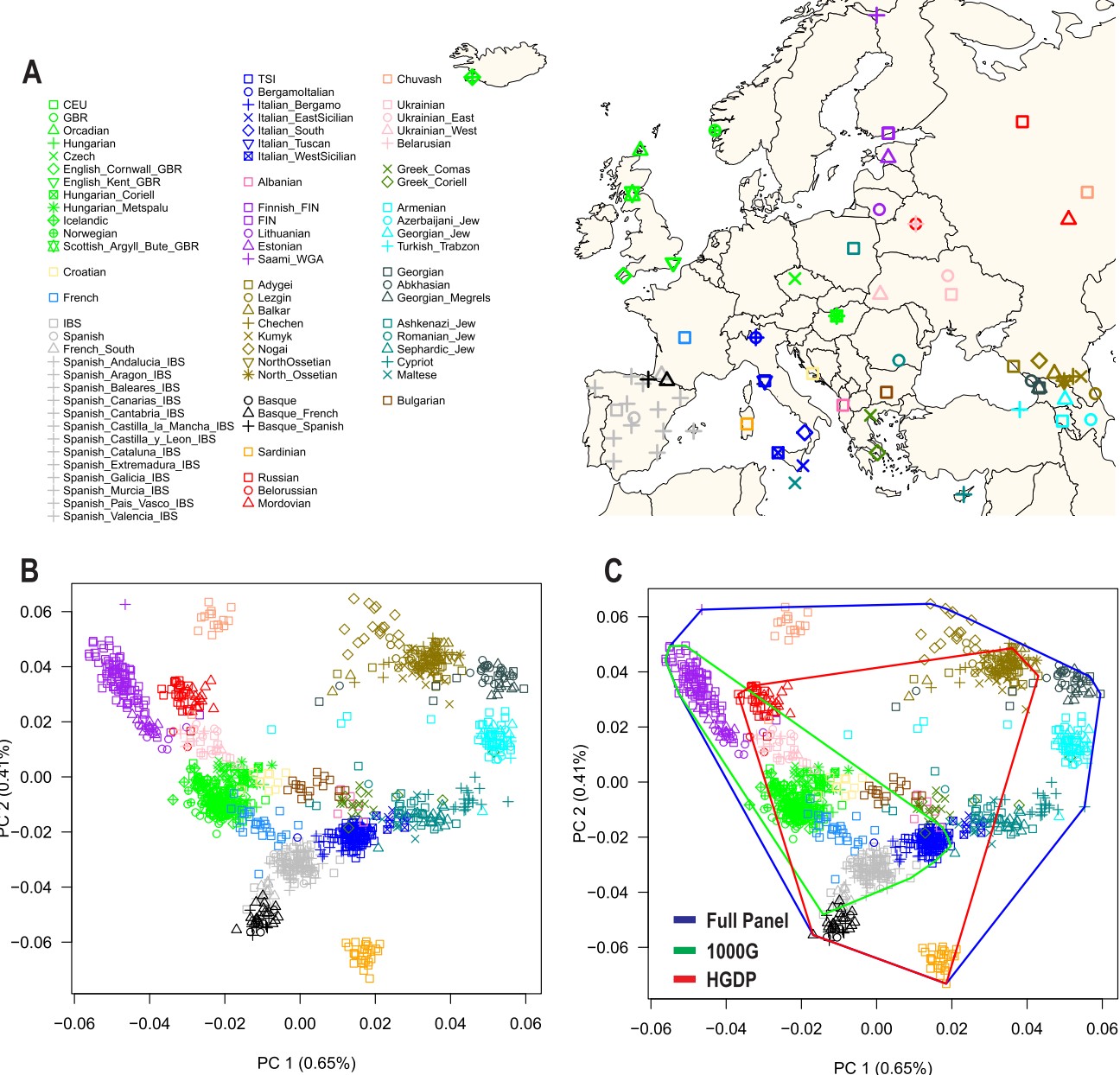

**Fig. 1 | European reference panels and coverage of European genetic diversity. A)** Map of Europe showing the geographic location of samples from 79 European populations. The map was drawn using the R package "maps" version 3.4.1. **B)** The first two principal components (PC1 and PC2) of genetic diversity and the percent variance explained. **C)** Coverage of genetic diversity over the first two principal components (convex hull area). 1000G = 1000 Genomes Project, HGDP Human Genome Diversity Project.

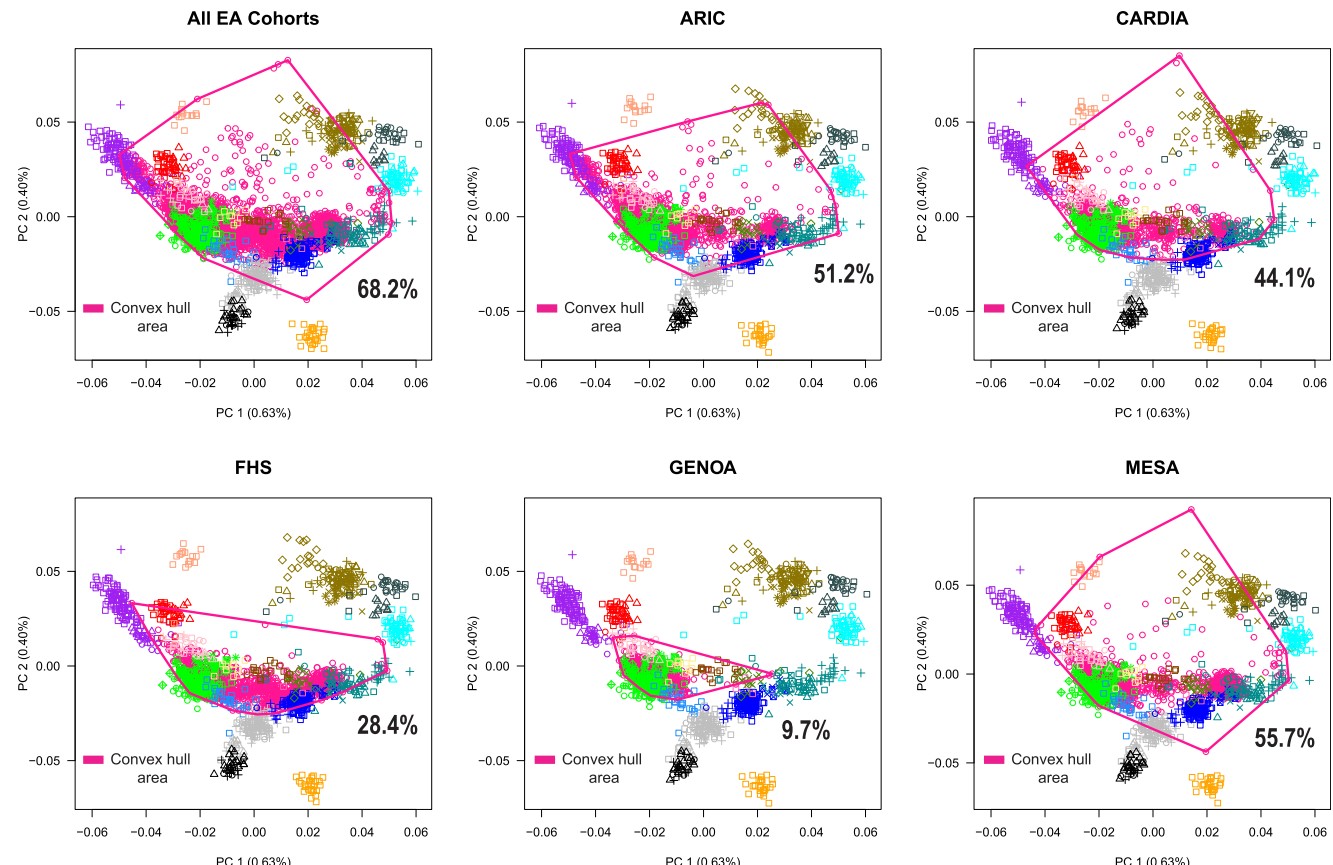

**Fig. 2 | Projection analysis of European Americans onto our European reference panel.** We plotted the convex hull area for all cohorts combined and for each European American cohort. The full legend as well as the geographic location of samples from 79 European populations can be found in Fig. 1. Convex hull area = Coverage of genetic diversity over the first two principal components. European Americans cohorts in the US: Atherosclerosis Risk in Communities (ARIC), Coronary Artery Risk Development in Young Adults (CARDIA), Framingham Heart Study (FHS), Genetic Epidemiology Network of Arteriopathy (GENOA), and Multi-Ethnic Study of Atherosclerosis (MESA).

comparable to $F_{ST}$ between British (GBR) and either Mexican (MXL, which have ~50% Native American ancestry, $F_{ST}$ = 0.031) or Punjabi in Pakistan (PJL, who have >70% South Asia Ancestry, $F_{ST}$ = 0.027) samples (Supplementary Data 3). Additionally, $F_{ST}$ estimates between European ancestry clusters are at least three-fold higher than $F_{ST}$ between Mandenka from Gambia in West Africa and Luhya from Kenya from East Africa ($F_{ST}$ = 0.011, Supplementary Data 3). Even when comparing real-world European populations, $F_{ST}$ estimates between Finnish in North Europe and Armenians or Georgians in South Europe are approximately twofold higher ($F_{ST}$ ~ 0.02) than $F_{ST}$ between Mandenka and Luhya ($F_{ST}$ = 0.011), i.e., between West and East Africans, and not as high as $F_{ST}$ between inferred European ancestry clusters. The $F_{ST}$ estimates in our analysis were in agreement with $F_{ST}$ values reported by the 1000 Genomes Project[35].

Supervised ADMIXTURE[17] analysis of European Americans (using individuals with ≥90% of one of three geography-associated European ancestry clusters as parentals) showed patterns of European ancestry clusters that differed by cohort (Fig. 4 and Supplementary Data 4). GENOA had the highest mean proportion of the North European ancestry cluster (44%, SE = 3.9%) and the lowest proportion of the Southeast European ancestry cluster (7%, SE = 3%), while FHS had the lowest mean proportion of the North European ancestry cluster (29.9%, SE = 3.7%). MESA had the highest proportion of the Southeast European ancestry cluster (25.4%, SE = 3.1%), followed by FHS (19.7% SE = 3%). The admixture patterns in the European American cohorts were consistent with the projection analysis (Fig. 2), e.g., the GENOA individuals clustered tightly with British and Scandinavian individuals on the first principal component. By testing genetic admixture using $f_3$

statistics[41], using Europeans as admixture sources and European Americans as admixture targets, we obtained formal evidence for admixture in the history of European Americans (Supplementary Data 5A–E). Also, we observed positive correlation between $F_{ST}$ (a measurement of North-South European differentiation) and $F_{IT}$ (a measurement of inbreeding) at SNPs throughout the genome in European American cohorts, consistent with subcontinental ancestry-related assortative mating in European Americans (Supplementary Data 6). Our results confirm the presence of subcontinental population structure in both Europeans and European Americans, that this structure reflects mixed ancestry in the vast majority of individuals, and that mixed ancestry reflects admixture rather than discrete subpopulations in Europe.

**Admixture dating in European Americans**

To date admixture in European Americans, we first applied a clustering approach[42] to the first two principal components and inferred that European Americans likely cluster within three subgroups of individuals (Fig. 5A and Fig. S3). Projection analysis of European Americans onto our European reference panel revealed that European Americans were widely distributed across a north-south axis, with centroids of inferred subgroups related to North (Subgroup N), Southwest (Subgroup SW), and Southeast (Subgroup SE) Europeans (Fig. 5B). The highest proportion of ancestry in Subgroup N individuals was North European ancestry (54.5%). Similarly, the highest proportions of ancestry in Subgroup SW and Subgroup SE individuals were Southwest European ancestry (53.7%) and Southeast European ancestry (71.2%), respectively. Next, we used MALDER[43] to infer admixture times for

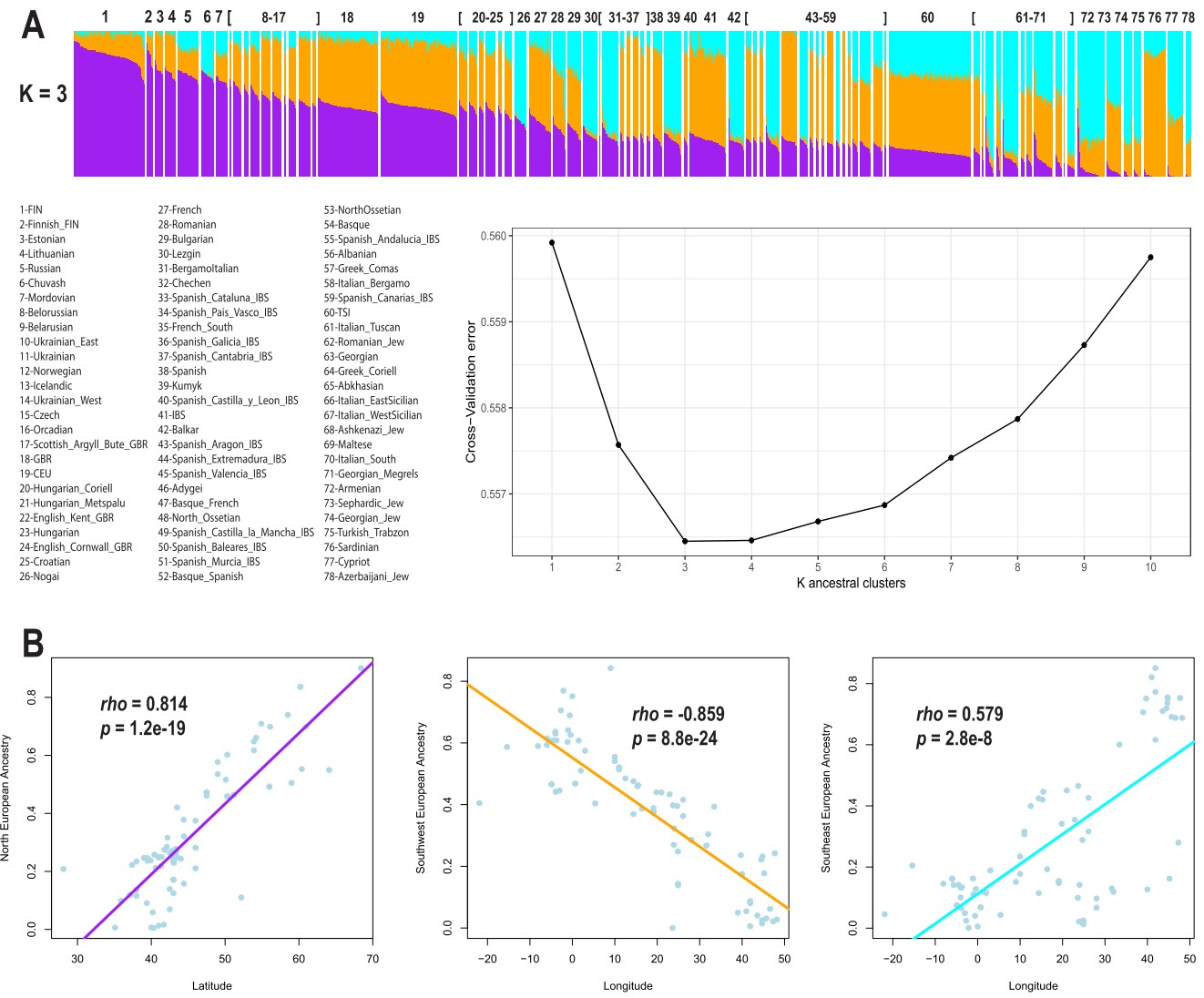

**Fig. 3 | Subcontinental ancestries in Europe and correlation of ancestry with geography. A)** Bar plot showing ancestry proportions in the European populations and a cross-validation plot supporting *K* = 3 as the most likely number of ancestry clusters. Purple, orange, and cyan colors represent ancestry clusters associated with North, Southwest, and Southeast European populations, respectively. Individual Bar plots were sorted in descending order of the amount of North European ancestry (Purple), and populations are sorted in descending order of the average of North European ancestry. **B)** Correlation plots depicting Spearman's *rho* between ancestry proportions and geographic coordinates. Colored lines represent fitted linear regressions. *p* value was derived from a two-sided test.

individuals within each of the three subgroups of European Americans. We observed significant admixture dating for all three subgroups, with subgroup SE yielding an admixture date ~10 generations more recent (42.00 generations, SE = 6.82) than admixture dates for subgroup SW (54.28 generations, SE = 10.43) and subgroup N (50.89 generations, SE = 14.26). As a confirmatory analysis, we used LaNeta[44] for admixture dating in European Americans, and we observed similar admixture times as inferred by MALDER (Supplementary Data 7).

### Implications of subcontinental admixture for association analysis

To understand the impact of subcontinental admixture in association studies and approaches to correct potential confounding, we investigated the classical association between *LCT* (rs4988235) and height, which has been claimed to be a false positive result due to stratification[27]. In addition, we evaluated the associations of rs4988235 with BMI and LDL, which were recently identified in large GWAS meta-analyses using primarily European-ancestry individuals (up to 500K samples)[14,25,26]. These studies either adjusted association models for genome-wide ancestry using the first 10 principal components[26] or

there was no evidence of adjustment for European population stratification[25]. Using our integrated set of European American cohorts (Supplementary Data 2), we replicated the previously reported associations between rs4988235 and height, LDL, and BMI when models were not adjusted for principal components, i.e., genome-wide ancestry (Fig. 6 and Supplementary Data 8). Different levels of adjustment for population structure (the genetic relatedness matrix, genome-wide ancestry [PCs], and/or locus-specific posterior probabilities of subcontinental European ancestry) attenuated the associations of rs4988235 with height and LDL (Fig. 6A, B and Supplementary Data 8). Importantly, when models were fully adjusted for both genome-wide and locus-specific subcontinental European ancestry, the associations of rs4988235 with height and LDL were completely eliminated, indicating that the unadjusted associations were false positives. In contrast, the association between rs4988235 and BMI remained weakly significant after adjustment for both genome-wide and locus-specific ancestry (Fig. 6C and Supplementary Data 8).

We also performed cohort-specific association analysis between rs4988235 and height, LDL, and BMI, (Supplementary Data 9–11).

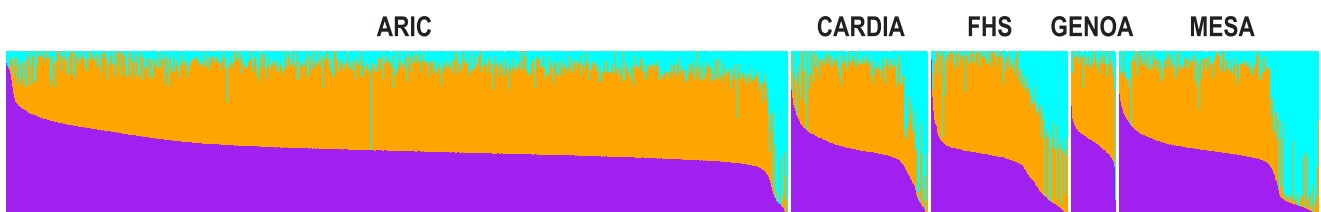

**Fig. 4 | Ancestry proportions in European Americans.** Bar plot representation of individual ancestry proportions inferred from supervised analysis. Purple, orange, and cyan colors represent ancestry clusters associated with North, Southwest, and Southeast European populations, respectively. Individual Bar plots were sorted in descending order of the amount of North ancestry cluster (Purple). European

Americans cohorts in the US: Atherosclerosis Risk in Communities (ARIC), Coronary Artery Risk Development in Young Adults (CARDIA), Framingham Heart Study (FHS), Genetic Epidemiology Network of Arteriopathy (GENOA), and Multi-Ethnic Study of Atherosclerosis (MESA).

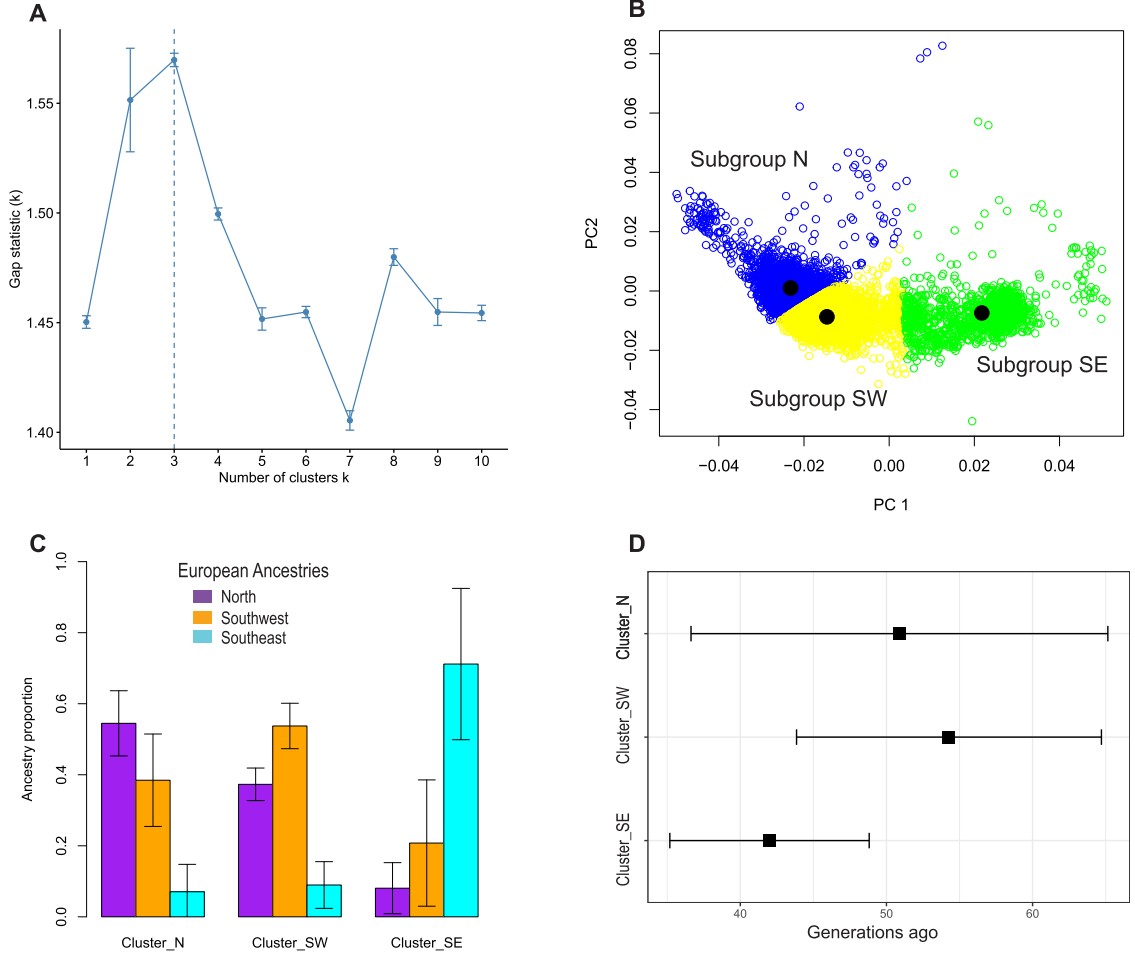

**Fig. 5 | Substructure and admixture dating in European Americans. A)** The number of clusters (*k*) was estimated using gap statistics (the error bars represent 95% confidence intervals), based on the first two principal components (PCs) derived from the **B)** projection analysis of European Americans (15,917 unrelated individuals). We estimated that European Americans are distributed across three subgroups representing North (N), Southwest (SW), and Southeast (SE) Europeans.

**C)** Bar plot representing ancestry profiles within each estimated cluster of European Americans. The error bars represent standard deviation. **D)** Admixture dating across clusters of European Americans. Point estimates and standard errors (represented by the error bars) of statistically significant admixture dates are shown on the horizontal axis.

When models were not adjusted for population stratification, the association between rs4988235 and height was significant in ARIC, CARDIA, FHS, and MESA but not in GENOA (Supplementary Data 9). The lack of association in GENOA might be explained by a small amount of ancestral heterogeneity and/or by small sample size. After adjustment for genome-wide ancestry, we observed association between rs4988235 and height in CARDIA but not in the other four cohorts. After adjustment for genome-wide and locus-specific

ancestry, we observed no association between rs4988235 and height in all five European American cohorts (Supplementary Data 9). Similarly for LDL, we observed some cohort-specific associations when models were not fully corrected, and that full adjustment attenuated or eliminated significance in all cohorts (Supplementary Data 10). These results imply that full ancestry adjustment (genome-wide and locus-specific subcontinental ancestry) may facilitate correction for residual stratification and avoidance of false positives in single studies.

**A      LCT−rs4988235−A and Height (n = 14,499)**

| Models | P−value |
|---|---|
| M1 = Height ~ Age + Sex | 5.328E−21 |
| M2 = M1 + [GRM] | 0.005 |
| M3 = M2 + Ref_PCs | 0.109 |
| M4 = M2 + Pop_PCs | 0.241 |
| M5 = M4 + Local_Anc | 0.672 |

Effect (Beta, cm)

**B      LCT−rs4988235−A and LDL (n = 14,575)**

| Models | P−value |
|---|---|
| M1 = LDL ~ Age + Sex | 3.52E−04 |
| M2 = M1 + [GRM] | 0.007 |
| M3 = M2 + Ref_PCs | 0.033 |
| M4 = M2 + Pop_PCs | 0.071 |
| M5 = M4 + Local_Anc | 0.327 |

Effect (Beta, mg/dl)

**C      LCT−rs4988235−A and BMI (n = 14,826)**

| Models | P−value |
|---|---|
| M1 = BMI ~ Age + Sex | 0.016 |
| M2 = M1 + [GRM] | 0.005 |
| M3 = M2 + Ref_PCs | 0.002 |
| M4 = M2 + Pop_PCs | 8.21E−04 |
| M5 = M4 + Local_Anc | 0.016 |

Effect (Beta, kg/m$^2$)

**Fig. 6 | Forest plots showing the association between rs4988235 and multiple traits, accounting for different levels of control of population stratification. A)** Height, **B)** LDL, and **C)** BMI. Forest plots show β values (95% confidence intervals represented by the error bars) and two-sided *p* values derived from linear mixed models. GRM genetic relatedness matrix, Ref_PCs PCs derived from a projection of individuals onto an ancestral reference panel, Pop_PCs PCs derived from within-population unsupervised PCA analysis, Local_Anc locus-specific ancestry.

To evaluate if subcontinental admixture could affect associations at other genetic loci, we performed genome-wide association analysis (GWAS) of height using models unadjusted and adjusted for genome-wide and locus-specific subcontinental ancestry (Fig. S4). As with rs4988235, we found that the associations with height for SNPs that were highly differentiated within Europe[29] were attenuated after adjustment for subcontinental admixture (Supplementary Data 12). Also, we identified 27 loci with signals of association with height in models unadjusted for locus-specific ancestry that were no longer statistically significant after adjustment for locus-specific ancestry (Supplementary Data 13 and Fig. S5). After interrogating the GWAS Catalog[45], none of these 27 loci were previously identified as associated with height. The additional adjustment for locus-specific ancestry in models accounting for genome-wide ancestry did not show a significant impact as assessed by genomic control (Fig. S4), but did control for confounding due to population stratification at genetic loci with locus-specific ancestry effects.

It is common practice in genetic association studies to account for genome-wide ancestry using principal components derived from study-specific unsupervised analysis (population-specific PCA). Here, we tested the approach of deriving principal components from projection of target individuals onto an external reference panel (projection or supervised PCA). To evaluate the similarity between these two approaches using our European American data, we performed Mantel's correlation test between individuals' genetic distances computed from the top twenty principal components obtained from the unsupervised and projection approaches. We observed moderate correlation in four studies (Mantel's *rho* from 0.46 to 0.53, *p* < 0.001), with GENOA not showing a significant correlation (Supplementary Data 14). Differences between these two PCA approaches may have led to differences in how well confounding was controlled. During testing of the association between rs4988235 and height, we observed better model fits (ΔAIC up to 12.45)[46] for some cohorts when models were adjusted for projection-derived principal components compared to study-specific principal components (Supplementary Data 9). For the integrated data set, study-specific principal components yielded better model fits than projection-derived principal components (Supplementary Data 8).

## Discussion

The existence of subcontinental-level ancestries has been documented within Africa and Asia[4,47–49], yet the presence of subcontinental ancestries within Europe is not well appreciated. We compiled genome-wide genotype and sequence data from geographically diverse Europeans and European Americans to investigate subcontinental-level ancestries and admixture in European-ancestry individuals. We also explore the consequences of different strategies for addressing ancestry in genetic epidemiology studies. Our study has four major results, described below.

First, we created a new reference panel of European genetic diversity by combining five genome-wide data sets[35–39]. We showed that panels based on the 1000 Genomes Project and the Human Genome Diversity Project, separate or combined, provided incomplete coverage of genetic diversity among Europeans or the European component of European Americans compared to our new reference panel. To facilitate genome-wide ancestry estimates, we provide as a research resource a reference SNP matrix of subcontinental ancestry-specific allele frequencies (https://github.com/mateushg1/CRGGH/). This resource allows for estimation of subcontinental ancestry proportions by projection analysis based on publicly available, aggregated, and non-identifiable data. The end-user does not need to access, clean, integrate, or analyze individual-level reference data. Additionally, we made available a detailed tutorial for performing ADMIXTURE and PCA projection analyses and using locus-specific ancestry posterior probabilities as covariates in GWAS analysis using PLINK 2.3[50].

Second, our admixture analyses yielded formal evidence that European-ancestry individuals are admixed at the subcontinental level. Moreover, our results support the occurrence of subcontinental ancestry-related assortative mating as a social factor that shaped the genetic structure of European Americans in the US[51]. Using multiple approaches to infer admixture, we showed that European-ancestry individuals are three-way admixed with wide variation in ancestry proportions. The demonstration that European Americans are ancestrally heterogeneous may have implications for calibrating locus-specific ancestry analysis[19] with respect to the number of generations since admixture began. Most recent admixture dates estimated for European Americans corresponded to the large-scale Migration Period in Europe (300–800 AD)[52], and were consistent with gene flow after the end of Roman Empire described in ancient DNA studies of the Viking Age[11] and Anglo-Saxon migrations[12]. A limitation of our study is that current methods for dating admixture have a limit of resolution of approximately 100 generations and tend to be biased toward more

recent admixture events. Another limitation of our study is a lack of ancestrally homogeneous reference populations or individuals corresponding to the Southeast European ancestry component.

Previous work has described projection analyses of ancient DNA samples in terms of present-day ancestries[53]. Southeastern European ancestry mainly represents descent from early Neolithic farmers from Anatolia who carried predominantly Y chromosome haplogroup G2a. Southwestern European ancestry mainly represents descent from Early to Middle Bronze Age southern steppe peoples (north and east of the Black Sea) who carried predominantly Y chromosome haplogroup R1b. Northern European ancestry mainly represents descent from Late Bronze Age northern steppe peoples (north and east of the Caspian Sea) who carried predominantly Y chromosome haplogroup R1a. Other ancestries not of European origin, such as Arabian, North African, and North Asian ancestries, have contributed to lesser extents to present-day Europe. Additionally, we have reconstructed the phylogeny of present-day ancestries[47]. One key inference from that reconstruction is that the ancestry reflected in early Neolithic farmers from Anatolia is likely the most recent common ancestor of present-day Southwestern European and Southeastern European as well as Arabian and North African ancestries.

Third, studies of European-ancestry individuals have reported that genetic variants, principally rs4988235, in the lactase gene (*LCT*) are associated with height, BMI, and LDL[25,26,54]. However, the association between rs4988235 and height has been suggested to be spurious due to uncorrected genome-wide ancestry[27]. Adjustment for genome-wide ancestry may not be sufficient to avoid false positive results and can mask true associations if ancestry is associated with the outcome[55]. Consistent with known potential confounding effects of ancestry[3,56], we demonstrated that the lack of adjustment for both genome-wide and locus-specific ancestry can produce false positives in association studies using European-ancestry individuals. By adjusting our models for locus-specific ancestry in addition to genome-wide ancestry, associations of rs4988235 with height and LDL were eliminated. In contrast, the association between rs4988235 and BMI remained after correcting for both genome-wide and locus-specific ancestry, suggesting an effect on weight but not on height. These results suggest that residual confounding by subcontinental European ancestry can produce spurious associations in genetic association studies, with consequences for estimation of polygenic adaptation[15] and polygenic risk scores[16] and for fine-mapping of genetic associations. Importantly, our results warrant further analyses on the impact of unmodeled European admixture on GWAS, polygenic adaptation, and polygenic risk scores in European-ancestry individuals, including those in large biobanks such as the UK Biobank[32] in Europe and the All of Us Research Program[33] and the VA Million Veteran Program[34] in the United States.

Fourth, for small studies, we tended to observe better model fit with adjustment for principal components derived from supervised analysis based on a common reference panel rather than for principal components derived from study-specific unsupervised analyses. However, the performance of unsupervised analysis approached or exceeded the performance of supervised analysis as the genetic diversity covered by the sample data approached or exceeded the genetic diversity covered by the external reference panel. European genetic diversity in our full panel covered by European American cohorts ranged from 9.7% to 55.7% whereas coverage reached 68.2% when all cohorts were combined. This result indicates that GWAS meta-analyses in which individual-level data cannot be or are not shared across studies should consider supervised analysis given a common reference. This recommendation does not depend on sample size, as even data sets on the scale of large biobanks do not necessarily cover a large proportion of ancestral diversity.

In conclusion, we demonstrated that European-ancestry individuals are admixed at the subcontinental level. Subcontinental admixture in Europeans and European Americans, if not properly accounted

for, can produce false positive associations in genetic epidemiology studies due to incomplete correction for confounding by ancestry. Our study highlights the need for full control, at both genome-wide and locus-specific ancestry levels, for confounding in Europeans and European Americans. Potential consequences of residual confounding by subcontinental ancestry include the misestimation of polygenic adaptation and poor performance of genetic or polygenic risk scores.

## Methods

### Samples
We compiled genome-wide data from five different studies: the 1000 Genomes Project[35], the Human Genome Diversity Project (HGDP)[36], the Human Origins dataset[37], a study of the Caucasus Mountains[38], and a study of the Jewish Diaspora[39] (Fig. 1A and Supplementary Data 1). Using these data, we created a data set that included 4796 individuals (worldwide reference panel), from which we extracted 1216 individuals from 79 European populations (European reference panel). We analyzed genome-wide array and phenotypic data from 17,684 European Americans from five genetic epidemiology cohorts, for which access was granted through dbGaP[57]: ARIC (phs000090.v1.p1), CARDIA (phs000285.v3.p2), FHS (phs000007.v32.p13), GENOA (phs000379.v1.p1), and MESA (phs000209.v13.p3).

### Data curation
To reduce batch effects due to the integration of array-based genotype data and whole genome sequence data, we performed quality control analysis within and between datasets using PLINK 1.9, filtering by minor allele frequency (--*maf 0.01*), per genotype missingness (--*geno 0.05*), per individual missingness (--*mind 0.05*), and deviation from Hardy Weinberg equilibrium (--*hwe 1 × 10⁻⁶*). We also pruned strand-ambiguous SNPs and SNPs in high linkage disequilibrium (--*indep-pairwise 50 10 0.8*).

### Population structure and relatedness
We used PLINK 1.9 to estimate the probability that individuals $i$ and $j$ share 0, 1, or 2 alleles identical by descent (IBD) ($\delta^0_{ij}$, $\delta^1_{ij}$, and $\delta^2_{ij}$, respectively)[50]. Based on these IBD probabilities, we calculated the pairwise kinship coefficient ($\Phi_{ij}$) as a function of IBD-sharing, $\Phi_{ij} = 1/2\delta^2_{ij} + 1/4\delta^1_{ij}$. We modeled the genetic relationships among individuals as networks[58], in which pairs of individuals were linked if they had a $\Phi_{ij}$ threshold ≥0.0884 (i.e., first- and second-degree relatives[59]). Then, we excluded related individuals using the maximum clique graph approach to minimize sample loss[58]. We performed unsupervised principal components analysis[13] and unsupervised ADMIXTURE analysis[17] on the European reference data. We performed unsupervised and supervised PCA and ADMIXTURE analyses using the reference data combined with the European American data. For supervised analysis in ADMIXTURE, we used as the ancestral references the European individuals with ≥90% of one of three ancestries based on unsupervised ADMIXTURE analysis. To evaluate the coverage of European diversity, we used the first two principal components to calculate convex hull areas[40]. We calculated $f_3$ statistics as implemented in ADMIXTOOLS[41] to formally test admixture. We tested all possible combinations of two European sources and a target European American cohort, following the form $f_3$(EUR_POP_X, EUR_POP_Y; EA_Cohort). All $f_3$ statistics with $z \leq -3$ were considered significant evidence of admixture. We used the top 20 principal components from population-specific and projection PCA approaches to calculate Euclidean distances. Then, we compared the correlation between the genetic distance matrices using Mantel's test implemented in the R package vegan[60].

### Admixture dating
We first combined all European American cohorts and performed supervised PCA by projecting the European Americans onto the

European reference panel. We then used gap and elbow statistics[42] to calculate the most likely number of clusters. To estimate the origin dates of admixture events, we calculated two-locus weighted LD decay statistics using MALDER[43] within each cluster of European Americans. Given that background LD can have a confounding effect on the weighted LD curves, we used as reference populations North European (Lithuanian and Estonian) and South European (Cyprus, Azerbaijani Jew, and Georgian Jew) populations that did not show high LD correlation with the tested target populations. For confirmatory analysis for admixture dating in European Americans, we calculated three-locus weighted LD decay using LaNeta[44]. Because LaNeta requires larger reference sample sizes to fit LD decay curves than MALDER, we included additional North European (Finnish in Finland [FIN]) and South European (Tuscans from Italy [TSI]) reference populations.

### Phasing and imputation
To generate valid VCF files before phasing, imputation, and association tests, we checked and corrected for monomorphic sites, consistency of reference alleles with the reference genome, variants with invalid genotypes, and non-SNP sites using the checkVCF.py Python script (https://github.com/zhanxw/checkVCF). We phased and imputed the genotype data using EAGLE2.4[61] and Minimac[62], respectively, using the TOPMed panel available through the TOPMed imputation server[63]. After imputation, we retained high quality SNPs with minor allele frequency ≥0.01 and with either high imputation quality (info ≥0.95) or empirically determined genotype data.

### Locus-specific ancestry analysis
Given that rs4988235 is highly differentiated between North and South European populations[30] and varies following a north-to-south gradient[27], we inferred two-way locus-specific ancestry using RFMix (version 1.5.4)[19]. Locus-specific ancestry estimates were performed using high quality imputed data. For ancestral references, we selected individuals with ≥90% North or South European ancestry as estimated in the unsupervised ADMIXTURE analysis. We performed inference in the PopPhased mode to correct possible phase errors. We set the number of generations since the admixture event (argument -G) at 50, the number of expectation maximization (EM) iterations (argument -e) at 2, and the window size (argument -w) at 0.2 cM. All other arguments were set at default values.

### Association analysis
To perform association analyses between rs4988235 and height, LDL, and BMI, accounting for different levels of control of population stratification, we used linear mixed models implemented in GENESIS[64]. Our analyses were focused on unrelated European Americans, with relatedness determined by the maximum clique graph approach[58]. Models were adjusted for the genetic relationship matrix as a random effect and the four first principal components (PCs that were significantly associated with the outcome and explained between-population structure) and/or locus-specific ancestry as fixed effects. Genome-wide ancestry was accounted for using principal components derived from one of two approaches: study-specific unsupervised analysis or supervised (projection) analysis of individuals onto an external reference panel. To account for the uncertainty of locus-specific ancestry estimates, models were adjusted for locus-specific ancestry dosages calculated from the posterior probabilities of locus-specific ancestry. Similarly, we used genotype dosages to account for imputation uncertainty. We performed genome-wide association analysis (GWAS) of height using models unadjusted and adjusted for locus-specific ancestry using PLINK 2.3, which allows for the inclusion of SNP-specific covariates. For the GWAS, we adjusted models for the top 12 PCs significantly associated with height.

### Ethics statement
All dbGaP studies (dbGaP Study Accession described in the "Methods" section) obtained ethical approval from the relevant institutions and written informed consent from each participant prior to participation. We obtained approval for controlled access (protocol number: 12-HG-N185) of each of the dbGaP studies.

### Reporting summary
Further information on research design is available in the Nature Portfolio Reporting Summary linked to this article.

## Data availability
The genome-wide and phenotypic data used in this manuscript are publicly available. Access to the European Americans from five genetic epidemiology cohorts was granted through dbGaP[57]: ARIC (phs000090.v1.p1), CARDIA (phs000285.v3.p2), FHS (phs000007.v32.p13), GENOA (phs000379.v1.p1), and MESA (phs000209.v13.p3). We provide as a research resource a reference SNP matrix of subcontinental ancestry-specific allele frequencies (https://github.com/mateushg1/CRGGH/). Publicly available data were retrieved from http://hgdownload.cse.ucsc.edu/gbdb/hg19/1000Genomes/phase3/, ftp://ngs.sanger.ac.uk/production/hgdp/hgdp_wgs.20190516/, https://reich.hms.harvard.edu/sites/reich.hms.harvard.edu/files/inline-files/EuropeFullyPublic.tar.gz, https://evolbio.ut.ee/caucasus/, and https://evolbio.ut.ee/jew/.

## Code availability
We have provided a pipeline on GitHub (https://github.com/mateushg1/CRGGH/)[65] for how to perform GWAS accounting for local ancestry, as well as how to perform ADMIXTURE and PCA projection analyses.

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

## Acknowledgements

This work utilized the computational resources of the NIH HPC Biowulf cluster (https://hpc.nih.gov). The contents of this publication are solely the responsibility of the authors and do not necessarily represent the official view of the National Institutes of Health. The Atherosclerosis Risk in Communities study has been funded in whole or in part with Federal funds from the National Heart, Lung, and Blood Institute, National Institutes of Health, Department of Health and Human Services, under contract numbers HHSN268201700001I, HHSN268201700002I, HHSN268201700003I, HHSN268201700004I, and HHSN268201700005I. The authors thank the staff and participants of the ARIC study for their important contributions. Funding for the GEN-EVA substudy was provided by National Human Genome Research Institute grant U01HG004402 (E. Boerwinkle). Support for the Coronary Artery Risk Development in Young Adults study was provided by NHLBI grant numbers HHSN268201300025C, HHSN268201300026C, HHSN268201300027C, HHSN268201300028C, and HHSN268201300029C (C. E. Lewis, D. Lloyd-Jones, P. Schreiner, S. Sidney, and J. Shikany). The Framingham Heart Study is conducted and supported by the National Heart, Lung, and Blood Institute (NHLBI) in collaboration with Boston University (contracts N01-HC-25195, HHSN268201500001I, and 75N92019D00031). This manuscript was not prepared in collaboration with investigators of the Framingham Heart Study and does not necessarily reflect the opinions or views of the Framingham Heart Study, Boston University, or NHLBI. Funding to support the Omni cohort recruitment, retention, and examination was provided by NHLBI contracts N01-HC-25195, HHSN268201500001I, and 75N92019D00031, as well as NHLBI grants R01-HL070100, R01-HL076784, R01-HL49869, and U01-HL-053941. SHARe Illumina genotyping was provided under an agreement between Illumina and Boston University. Support for GENOA was provided by the National Heart, Lung and Blood Institute (HL054457, HL054464, HL054481, HL119443, and HL087660) of the National Institutes of Health. We would like the thank the Mayo Clinic Genotyping Core, the DNA Sequencing and Gene Analysis Center at the University of Washington, and the Broad Institute for their genotyping and sequencing services. We would like to thank the GENOA participants. This manuscript was not prepared in collaboration with investigators from the Genetic Epidemiology Network of Arteriopathy and does not necessarily reflect the opinions or views of the Genetic Epidemiology Network of Arteriopathy or NHLBI. Funding for CARe genotyping was provided by NHLBI Contract N01-HC-65226. MESA and the MESA SHARe project are conducted and supported by the National Heart, Lung, and Blood Institute (NHLBI) in collaboration with MESA investigators. Support for MESA is provided by contracts N01-HC95159, N01-HC-95160, N01-HC-95161, N01-HC-95162, N01-HC-95163, N01-HC-95164, N01-HC-95165, N01-HC95166, N01-HC-95167, N01-HC-95168, N01-HC-95169, UL1-RR-025005, and UL1-TR-000040. Funding for SHARe genotyping was provided by NHLBI Contract N02-HL-64278. Genotyping was performed at Affymetrix (Santa Clara, California, USA) and the Broad Institute of Harvard and MIT (Boston, MA, USA) using the Affymetrix Genome-Wide Human SNP Array 6.0. This manuscript was not prepared in collaboration with MESA investigators and does not necessarily reflect the opinions or views of MESA, or the NHLBI. The study was supported in part by the Intramural Research Program of the National Institutes of Health in the Center for Research on Genomics and Global Health (CRGGH). The CRGGH is supported by the National Human Genome Research Institute, the National Institute of Diabetes and Digestive and Kidney Diseases and the Office of the Director at the National Institutes of Health (1ZIAHG200362). E.T.-S. was funded by CNPq-Brazil (Conselho Nacional de Desenvolvimento Científico e Tecnológico) and FAPEMIG (Minas Gerais State research Agency). The funders had no role in the design and conduct of the study; collection, management, analysis, and interpretation of the data; preparation, review, or approval of the manuscript; and decision to submit the manuscript for publication.

## Author contributions

The project was conceived by M.H.G., D.S., C.N.R., and A.A.A. M.H.G. and D.S. assembled datasets. M.H.G., T.P.L., and D.S. analyzed genetic data. M.H.G., D.S., A.R.B., E.T.-S., C.D.B., A.A.A., and C.N.R. contributed to data interpretation. M.H.G., D.S., A.A.A., and C.N.R. wrote the manuscript. All authors read the manuscripts and contributed with suggestions.

## Funding

## Competing interests

The authors declare no competing interests.
