## [Peer Review File · Nature Communications]

Unappreciated Subcontinental Admixture in Europeans and European Americans and Implications for Genetic Epidemiology StudiesREVIEWER COMMENTS

Reviewer #1 (Remarks to the Author):

This study provides a comprehensive reference panel of European genetic diversity, which far exceeds the ancestry coverage provided by the widely used 1KGP and HGDP resources. Using this panel, together with five different European American cohorts, the authors provide a completely convincing argument for admixture among European populations, with three distinct subcontinental ancestry components. They go on to show how controlling for genome-wide and local subcontinental ancestry can refine genotype-phenotype association estimates for European ancestry cohorts. They make a strong suggestion that their favored approach of using projected PC values, as opposed cohort-specific PC values, and local ancestry as controls for ancestry/population structure should be adopted as best practices for association studies in European ancestry cohorts.

This manuscript was a pleasure to read and review. It is very clearly written and presented, represents an important contribution to the field, and appears to be largely technically sound. I have two main concerns regarding the analysis and interpretation of the results, which I elaborate on in points #1 and #2 below. I also provide suggestions for how the authors can support the adoption of their approach by the research community, and finally I enumerate a number of other relatively minor questions and suggestions.

Major comments

1. Technical: my only substantive technical concern relates to the F_{st} estimates, which do not seem to accurately capture the genetic differentiation among the ancestry groups analyzed here. It is incomprehensible that F_{st} between North and Southwest European groups 0.032 is greater than the F_{st} for GBR and MXL 0.031, given that the latter comparison also includes and North vs. Southwest European comparison along with a comparison distinct European and Native American continental groups. The lower F_{st} value of 0.027 for GBR and PJI is even more inexplicable. Similarly, the 3x lower F_{st} of 0.011 for Mandenka and the East African Bantu Luhya is difficult to reconcile with the known population history of these groups. I suspect that these results reflect the inadequacy of the univariate approach to F_{st} , averaging across loci genome-wide, which works well for single loci but not for quantifying genomic divergence. It would be interesting compare between group distances for these groups using multivariate methods like GRM or PCA distances (as the authors describe on page 8). Presumably, there would be greater genetic differentiation between than within continental regions.

2a. Conceptual: a major conclusion of the manuscript is that adjustment for subcontinental European ancestry at both genome-wide and locus-specific levels should be adopted as best practices for genetic association studies using European ancestry individuals. I am not convinced that they have made the case for the necessity of adjustment by locus-specific ancestry. For example, the confidence intervals for M3 (Ref_PCs), M4 (Pop_PCs), and M5 (Local_Anc) all substantially overlap for all three comparisons in Figure 6. M4 and M5 both attenuate the significant associations between rs4988235, Height and LCL. It seems that adjustment for PCs alone is sufficient.

2b. Projected vs. population-specific PCs: similarly, the M4 Pop_PCs show more attenuation of the association signals than the M3 Ref_PCs in all three comparisons, although the confidence intervals overlap substantially.

2c. Genome-wide ancestry: Why were only the first 4 PCs used to control for population structure (genome-wide ancestry) in the association analysis? If more PCs are used, does it attenuate the associations between rs4988235, Height and LCL more, thereby obviating the need for locus-specific ancestry adjustment?

2d. Local ancestry: it is not clear why two-way local ancestry inference was used given the clear three-way admixture patterns seen for European populations. The methods section states that this was done to accommodate the north-south ancestry gradient in Europe, but all European reference populations, and all European ancestry individuals analyzed here show three-way admixture. It is not clear how the third ancestry component (the remaining ~10% given their 90% cutoff) is modeled in this analysis.

3. Data and code availability: This paper advocates for changes to best practices for genetic

association studies in European ancestry cohorts – using PC values from projections onto their more comprehensive European ancestry reference panel and adjusting for local ancestry. To support this, they provide a reference SNP matrix of subcontinental ancestry-specific allele frequencies and scripts for running RFMix. Potentially interested adopters of their method are left to figure out on their own how to do the projection and how to include the local ancestry calls as controls in association studies. This may be OK, since these are widely used (if not completely standard) techniques in association studies. But they are likely to get much better adoption of their method if they were to provide a detailed tutorial for how to implement their suggested best practices into standard association study pipelines.

Minor comments

4. All PCAs show the percent variance explained as fractions. For example, in Figure 1, PC1 is shown to explain 0.65% of the variance. This is confusing. Does it really explain less than 1% of the total variance, or does it explain 65% of the variance (i.e. 0.65)? Is it possible that PC1 and PC2 together only explain ~1% of the variance for European reference populations?

5. In the introduction, several methods are cited for genome-wide ancestry but only a single method is cited for local ancestry (RFMix). The authors may consider citing newer methods for local ancestry inference, such as Gnomix and FLARE.

6. For the ancestry of the cohorts studied here, three PCA clusters are mentioned, including one overlapping Finnish individuals (page 5). But Figure 2 shows that the Finnish reference individuals (upper left, purple) do not fall within the convex hull areas for the cohorts.

7. If the authors could elaborate a bit on how the European admixture components related to ancient European admixture, that would be very interesting. From the discussion, I take that the admixture is more recent (300-800 AD), but it is not clear how the three European subcontinental ancestry components do, or do not, relate to the ancient populations that admixed to form modern Europeans.

8. Please include a description of how individuals' genetic distances were computed from the top twenty principal components (page 8) in the methods section.

Reviewer #2 (Remarks to the Author):

In this paper, Gouveia et al. investigated the ancestral diversity in Europeans and European Americans, pointing out differences in admixture dates between different populations and the impact of this admixture in association studies.

The major noteworthy observation is that subcontinental admixture in individuals with European ancestry if not properly taken into consideration, can produce spurious results in genetic epidemiology studies. The topic is definitely of great interest however, the study lacks several additional analyses to support the claims.

The first section of the article is all about admixture dating and admixture detection, the methods used are mainly allele frequencies-based, however, given how closely related all European populations are, haplotype-based methods are better and more precise. Thus, I suggest the use of Chromopainter and Finestructure in their data, and then the author should estimate admixture with Globetrotter. If knowing that the admixture level is important to avoid spurious association I think that, for European populations, haplotype-based methods are best suited to the task. I encourage the author to run the mentioned software on their data and compare the results.

Link for GLOBETROTTER

<https://people.maths.bris.ac.uk/~madjl/finestructure/globetrotter.html>

The second section shows the potential implications of confounding by subcontinental ancestry and admixture, which is the major take-home message of the article.

Unfortunately, all the analyses are based only on the association between rs4988235 and height, LDL-

cholesterol, and BMI. The study should include many more genes and variants linked to multiple phenotypes that show differentiation between North and South Europeans or under selection in European populations. Certainly, the LCT gene is not the only one where subcontinental ancestry and admixture have an impact, but the authors should at least give some information on this impact, how many genes could be affected? Could they include additional ten genes? The LCT gene is the only one that shows this pattern?

In addition, the authors mentioned the impact of unaccounted admixture on polygenic risk score in the abstract, but the study lacks any analyses on this aspect. As they did with LDL-cholesterol and BMI, the authors should show the usefulness of their reference panel in estimating polygenic risk scores. The authors should measure the impact of such admixture in the errors in estimating polygenic risk score. In this case, they could show the change in polygenic risk score with and without admixture and measure the error in the prediction.

I am also suggesting running extensive simulations using simulation tools such as SLIM about subcontinental admixture and measuring the subsequent impact on spurious associations, one interesting question is linked to the amount of cryptic gene flow needed to create a spurious association.

Overall, I think the topic is intriguing but more analyses are needed for this article to be published. The article as it is now it seems more like an interesting pilot study.

REVIEWER COMMENTS

Reviewer #1 (Remarks to the Author):

Comment 1. Technical: my only substantive technical concern relates to the F_{ST} estimates, which do not seem to accurately capture the genetic differentiation among the ancestry groups analyzed here. It is incomprehensible that F_{ST} between North and Southwest European groups 0.032 is greater than the F_{ST} for GBR and MXL 0.031, given that the latter comparison also includes and North vs. Southwest European comparison along with a comparison distinct European and Native American continental groups. The lower F_{ST} value of 0.027 for GBR and PJI is even more inexplicable. Similarly, the 3x lower F_{ST} of 0.011 for Mandenka and the East African Bantu Luhya is difficult to reconcile with the known population history of these groups. I suspect that these results reflect the inadequacy of the univariate approach to F_{ST} , averaging across loci genome-wide, which works well for single loci but not for quantifying genomic divergence. It would be interesting compare between group distances for these groups using multivariate methods like GRM or PCA distances (as the authors describe on page 8). Presumably, there would be greater genetic differentiation between than within continental regions.

Response: Thank you for your concern about the accuracy of F_{ST} in our analyses. We compared our F_{ST} estimates with those published by the 1000 Genomes Project (PMID: 26432245), which was based on high coverage whole-genome sequencing data. We confirmed that our F_{ST} analyses are accurate and in agreement with F_{ST} values reported by the 1000 Genomes Project (see examples below of F_{ST} for populations mentioned in the main text of our manuscript). We used the Weir and Cockerham F_{ST} estimator (PMID: 28563791), which is the recommended measure of population divergence. Additionally, it has been previously shown that there is no difference between the measure of population divergence based on F_{ST} or PCA distances (PMID: 19834557). An important point is that we calculated F_{ST} between European clusters, which are genetically homogeneous virtual populations identified by ADMIXTURE. To put the differentiation between European clusters into the context of real populations, we calculated F_{ST} between specific population groups in our data (e.g., GBR vs MXL and GWD vs LWK), which are admixed at both continental and subcontinental levels. The F_{ST} value of 0.011 between LWK and GWD is expected due to the recent common origin of these two populations (Bantu expansion, PMID: 28473590). Additionally, it should be noted that the 1000 Genomes Project had an explicit strategy of sampling closely related groups (“an efficient way to find variants is to sample a set of geographically related populations with about 1% F_{ST} differentiation among them” PMID: 26432245). The issue raised by the reviewer can be explained by noting the difference between the sampling origin (or residence) and the ancestral origin. These points were addressed in the revised version of our manuscript in the Results, page 6.

F_{ST} estimates		
Populations	1000 Genomes Project	Our study
GBR vs MXL	0.036	0.031
GBR vs PJI	0.028	0.027
GWD vs LWK	0.011	0.011

Comment 2A. Conceptual: a major conclusion of the manuscript is that adjustment for subcontinental European ancestry at both genome-wide and locus-specific levels should be adopted as best practices for genetic association studies using European ancestry individuals. I am not convinced that they have made the case for the necessity of adjustment by locus-specific ancestry. For example, the confidence intervals for M3 (Ref_PCs), M4 (Pop_PCs), and M5 (Local_Anc) all substantially overlap for all three

comparisons in Figure 6. M4 and M5 both attenuate the significant associations between rs4988235, Height and LCL. It seems that adjustment for PCs alone is sufficient.

Response: We thank the reviewer for this important comment questioning if only PC adjustment could be sufficient. Assessment of overlapping confidence intervals is not a valid statistical method to perform model selection. We performed model selection using the Akaike Information Criterion, which is based on penalized likelihoods. As shown in Supplementary Table S8, adjustment for locus-specific ancestry (M5 vs. M4) led to no improvement for height, an insignificant improvement for LDL, and a significant improvement for BMI. Whether adjustment for locus-specific ancestry yields improvement depends on the specific circumstances, but the best practice is to perform the adjustment.

To further investigate this issue, we performed genome-wide association analysis of height using models unadjusted and adjusted for local ancestry. We identified 27 loci associated with height in models unadjusted for local ancestry, but that lost significance after adjustment for local ancestry (new Table S13 and new Figures S4 and S5). These 27 loci represent the type of false positive errors or spurious associations for which adjustment by PCs alone is insufficient. We included a new paragraph with these results in the revised version of our manuscript in the Results, page 8.

Comment 2B. Projected vs. population-specific PCs: similarly, the M4 Pop_PCs show more attenuation of the association signals than the M3 Ref_PCs in all three comparisons, although the confidence intervals overlap substantially.

Response: For the integrated data set, the model fit for unsupervised analysis is better than the model fit for supervised analysis for all three phenotypes (Table S8). In contrast, for the cohort-specific analyses, supervised analysis yielded a better model fit than unsupervised analysis 53% of the time (Tables S9-S11). For the integrated data set, Ref_PC 1 and Pop_PC 1 are highly correlated with each other, with both explaining a north-south gradient. To the reviewer's point, unsupervised analysis may outperform supervised analysis if the reference panel does not contain an ancestry that is present in the sample. We see evidence supporting this possibility in Figures 2 and S2, with a small number of individuals in CARDIA and MESA plotting outside the convex hull area (Figure 2) or the centroid of the European cluster (Figure S2). Also, unsupervised analysis may outperform supervised analysis if the supervised analysis contains an ancestry absent from the sample. In this situation, supervised analysis will waste free parameters, yielding a worse penalized likelihood. We see evidence supporting this possibility in Figure 2, with no individuals in the European American studies plotting to the region of the Basque and Sardinian reference clusters along Ref_PC2. We therefore offer in the new version of our manuscript (in the Discussion, page 11) a more nuanced recommendation that both unsupervised and supervised analyses be evaluated to determine empirically which offers better control.

Comment 2C. Genome-wide ancestry: Why were only the first 4 PCs used to control for population structure (genome-wide ancestry) in the association analysis? If more PCs are used, does it attenuate the associations between rs4988235, Height and LCL more, thereby obviating the need for locus-specific ancestry adjustment?

Response: Thanks for this important comment. There are two conditions for a PC to be a confounder: the PC must explain a significant amount of variance in the genotype data and the PC must be associated with the outcome. It should be noted that adjustment for the random effect of the GRM is equivalent to simultaneous adjustment for all PCs, if none of the eigenvalues are exceptionally large. We used the top four PCs to control for population structure because they were significantly associated with the studied phenotypes and explained between population structure, rather than population-specific population

structure (Fig. S1). Following the reviewer's suggestion, models were adjusted for all 12 PCs significantly associated with height in our new GWAS analysis. Even in models adjusted for 12 PCs, local ancestry further attenuates the association and reduces significance for rs4988235. We observed the same behavior in new analysis using other variants highly differentiated in Europe (new Table S12). The 27 variants identified with significant local ancestry effect (new Table S13) were found in models adjusted for all significant PCs (12 PCs), illustrating that the adjustment for PCs is not sufficient to obviate the need for locus-specific ancestry adjustment.

Comment 2D. Local ancestry: it is not clear why two-way local ancestry inference was used given the clear three-way admixture patterns seen for European populations. The methods section states that this was done to accommodate the north-south ancestry gradient in Europe, but all European reference populations, and all European ancestry individuals analyzed here show three-way admixture. It is not clear how the third ancestry component (the remaining ~10% given their 90% cutoff) is modeled in this analysis.

Response: We appreciate the reviewer's comment. The local ancestry analysis is a supervised approach in which the reference populations are defined a priori, and more homogenous reference populations will provide more accurate local ancestry calls. We had individuals with > 90% North European ancestry or South European ancestry. One limitation of our study is that we did not have homogeneous reference individuals for the Southeast ancestral component. In our previous publication (Reference 42), we noted that Abkhasian and Georgian samples had the highest proportions of a Western Asian ancestral component, but these proportions did not exceed 60%. We continue to search for new data to address this key limitation. We acknowledged this limitation in the new version of our manuscript in the Discussion, page 10. Additionally, we adjusted the association models for local ancestry probabilities, thus the uncertainty of local ancestry calls is accounted for in our models.

Comment 3. Data and code availability: This paper advocates for changes to best practices for genetic association studies in European ancestry cohorts – using PC values from projections onto their more comprehensive European ancestry reference panel and adjusting for local ancestry. To support this, they provide a reference SNP matrix of subcontinental ancestry-specific allele frequencies and scripts for running RFMix. Potentially interested adopters of their method are left to figure out on their own how to do the projection and how to include the local ancestry calls as controls in association studies. This may be OK, since these are widely used (if not completely standard) techniques in association studies. But they are likely to get much better adoption of their method if they were to provide a detailed tutorial for how to implement their suggested best practices into standard association study pipelines.

Response: Thanks for this important suggestion. We have made available on GitHub a detailed tutorial on how to perform ADMIXTURE and PCA projection analyses and how to include local ancestry calls as per-locus covariates in association analysis using PLINK2 (<https://github.com/mateushg1/CRGGH/>). Specifically, we provided all command line code to perform the analyses and a Python script to extract RFMix local ancestry calls and format them for use with the --local-covar function. Mention of the availability of these new resources has been added to the new version of our manuscript in Discussion, page 9, and in the Data/Code Availability, page 15.

Minor comments

Comment 4. All PCAs show the percent variance explained as fractions. For example, in Figure 1, PC1 is shown to explain 0.65% of the variance. This is confusing. Does it really explain less than 1% of the total

variance, or does it explain 65% of the variance (i.e. 0.65)? Is it possible that PC1 and PC2 together only explain ~1% of the variance for European reference populations?

Response: *We understand the reviewer concern; however, our results are correct and consistent with previous studies of human genetic diversity. PC1 and PC2 of world-wide human populations explain from 9.1% and 2.7% of variance, respectively, while PC1 of Eurasians and PC2 of Eurasians explain 0.9% and 0.4%, respectively (PMID: 25230663). Another report evaluating European substructure showed that the two first principal components of European stratification explained 0.38% and 0.064% of variance (PMID: 26124090). It is not possible that PC1 explains 65% of the variance and PC2 explains 41% of the variance, as PC1 and PC2 together would explain >100% of the variance.*

Comment 5. In the introduction, several methods are cited for genome-wide ancestry but only a single method is cited for local ancestry (RFMix). The authors may consider citing newer methods for local ancestry inference, such as Gnomix and FLARE.

Response: *We thank the reviewer for the suggestions of citations. We agree that RFMix is dated, and we are aware of both Gnomix and FLARE. We have cited both recently published methods to infer local ancestry. To be clear, neither we nor the research community endorse either of these new methods at this time, as they have not been vetted by the research community yet.*

Comment 6. For the ancestry of the cohorts studied here, three PCA clusters are mentioned, including one overlapping Finnish individuals (page 5). But Figure 2 shows that the Finnish reference individuals (upper left, purple) do not fall within the convex hull areas for the cohorts.

Response: *We agree that Finnish reference individuals do not fall within the convex hull area for GENOA. Few Finnish reference individuals fall within the convex hull areas for CARDIA, FHS, and MESA. For the ARIC study, the convex hull area covers several Finnish reference individuals. We included this observation in the new version of our manuscript in the Results, page 5.*

Comment 7. If the authors could elaborate a bit on how the European admixture components related to ancient European admixture, that would be very interesting. From the discussion, I take that the admixture is more recent (300-800 AD), but it is not clear how the three European subcontinental ancestry components do, or do not, relate to the ancient populations that admixed to form modern Europeans.

Response: *Thanks for the comment. We added a discussion of previous analyses of ancient DNA in terms of present-day ancestries described in our current manuscript. In the context of Comment 1, it is important to consider that the three major ancient populations (southern steppe, northern steppe, and Anatolian early farmer) that contribute to the bulk of present-day European ancestry are not of European origin but rather are of Central or Western Asian origin. Upon recognition of a common ancestor of present-day Europeans and Native Americans and of a common ancestor of present-day Europeans and South Asians somewhere in Asia, as well as mass migration events leading to large-scale population turnover within Europe, it is easy to appreciate the limited utility of the concept of continental origins of present-day populations. We also clarified that our analyses estimated the most recent admixture dates, for the purpose of tuning RFMix locus-specific ancestry analysis in the new version of our manuscript in the Discussion, page 10.*

Comment 8. Please include a description of how individuals' genetic distances were computed from the top twenty principal components (page 8) in the methods section.

Response: *Thanks for the comment. We included in the Methods section (page 13) the description of how we calculated individuals' genetic distances using the top twenty principal components.*

Reviewer #2 (Remarks to the Author):

Comment 1. The first section of the article is all about admixture dating and admixture detection, the methods used are mainly allele frequencies-based, however, given how closely related all European populations are, haplotype-based methods are better and more precise. Thus, I suggest the use of Chromopainter and Finestructure in their data, and then the author should estimate admixture with Globetrotter. If knowing that the admixture level is important to avoid spurious association I think that, for European populations, haplotype-based methods are best suited to the task. I encourage the author to run the mentioned software on their data and compare the results.

Link for GLOBETROTTER

<https://people.maths.bris.ac.uk/~madjl/finestructure/globetrotter.html>

Response: *We thank the reviewer for the suggestion. While admixture detection is important in our manuscript, admixture dating is not. Specifically, the purpose of admixture dating is solely the tuning of RFMix local ancestry inferences. The authors of fineSTRUCTURE/ChromoPainter wrote that "if individuals i and j share a distinctive haplotype tract, then they will both be counted as donors for each other and the same chunk will appear in the likelihood twice, once in x_{ij} and the second in x_{ji} " (PMID: 22291602). Using the data twice in this manner violates the chain rule and leads to overfitting. RFMix is haplotype-based yet does not suffer from this problem. From a Bayesian perspective, the prior probability of recombination based on the combination of the genetic map and the dating parameter tends to be overwhelmed by the data, such that RFMix is not sensitive to misspecification of the dating parameter. RFMix also has functionality to correct phasing errors, thereby reducing or avoiding incorrect local ancestry estimates due to phasing artifacts that can adversely affect other haplotype-based methods. After reviewing methods for admixture dating, we note that ALDER (PMID: 23410830), MALDER (PMID: 24550290), GLOBETROTTER (PMID: 24531965), MOSAIC (PMID: 31123038), and fastGLOBETROTTER (PMID: 35794007) are based on two-locus LD. A recently published method, LaNeta (PMID: 358392492), is the only method that uses three-locus LD. We performed new analysis with LaNeta, and we confirmed admixture dates as estimated by MALDER (new Table S7).*

Comment 2. The second section shows the potential implications of confounding by subcontinental ancestry and admixture, which is the major take-home message of the article. Unfortunately, all the analyses are based only on the association between rs4988235 and height, LDL-cholesterol, and BMI. The study should include many more genes and variants linked to multiple phenotypes that show differentiation between North and South Europeans or under selection in European populations. Certainly, the LCT gene is not the only one where subcontinental ancestry and admixture have an impact, but the authors should at least give some information on this impact, how many genes could be affected? Could they include additional ten genes? The LCT gene is the only one that shows this pattern?

Response: *We thank the reviewer for this important comment. In addition to rs4988235, we performed association analysis of height for 32 other variants that are highly differentiated in Europe (new Table S11), and we showed that the pattern observed for rs4988235 also occurred for other several loci/genes.*

We included a new paragraph with these results in the revised version of our manuscript in the Results, page 8.

Comment 2. In addition, the authors mentioned the impact of unaccounted admixture on polygenic risk score in the abstract, but the study lacks any analyses on this aspect. As they did with LDL-cholesterol and BMI, the authors should show the usefulness of their reference panel in estimating polygenic risk scores. The authors should measure the impact of such admixture in the errors in estimating polygenic risk score. In this case, they could show the change in polygenic risk score with and without admixture and measure the error in the prediction.

Response: *We thank the reviewer for this comment. We elaborated on this topic as a future direction in the discussion of our current manuscript in the Discussion, page 11. We agree that the topic is important and timely, but the analysis is beyond the scope of the current manuscript. We are currently designing a project to evaluate the impact of unaccounted admixture on polygenic risk scores using large-scale biobank data.*

Comment 3. I am also suggesting running extensive simulations using simulation tools such as SLIM about subcontinental admixture and measuring the subsequent impact on spurious associations, one interesting question is linked to the amount of cryptic gene flow needed to create a spurious association.

Response: *We thank the reviewer for the comment. After careful consideration, we conclude that the question raised by the reviewer is not limited to the amount of gene flow required to create spurious association. Rather, the question is much broader, involving splitting of source populations, gene flow, intermating, the genetic architecture of the phenotype, and the sampling scheme. As such, the simulations are infinite. Specifically with respect to gene flow, there is a result known from population genetics theory. Letting m be the immigration rate per generation and N_e be the effective size of the recipient population, Wright (1931) established that there is population structure if m is less than $1/2N_e$ and there is panmixis if m is greater than $1/2N_e$ (PMID: 17246615). Given that this result is not new, we approached the comment from the pragmatic perspective of whether spurious associations have been detected or reported in analyses of empirical data. To address this point, we performed genome-wide association analysis of height using models unadjusted and adjusted for local ancestry. We identified 27 loci associated with height in models unadjusted for local ancestry but not associated after adjustment for local ancestry (new Table S13). These loci represent spurious associations avoided by our methodological approach. We included these new results in the revised version of our manuscript in the Results, page 8.*

REVIEWERS' COMMENTS

Reviewer #1 (Remarks to the Author):

The authors have satisfactorily addressed all of my concerns.

Reviewer #2 (Remarks to the Author):

The authors have satisfactorily addressed the questions I raised in my previous review.